# Dynamic Novel View Synthesis from Unsynchronized Videos using Global-Local Motion Consistency Prior

## Abstract

Dynamic novel view synthesis (D-NVS) critically depends on hardware-based synchronization. Current approaches that accommodate unsynchronized settings within the widely-used NeRF or GS frameworks often struggle with local minima, particularly in textureless scenes or when multi-view videos exhibit large misalignments. To tackle this issue, we propose a novel 3D global–2D local motion consistency prior, which evaluates the alignment between predicted scene flow projections and pre-computed optical flows across multi-view videos. Our analysis reveals that the motion, produced by the anisotropy of projected global scene flow across different views, is inherently more effective for correcting temporal misalignments compared to the near-isotropic appearance typically leveraged in NeRF or GS. Extensive experiments on public datasets demonstrate the versatility of our loss function across various D-NVS architectures (NeRF and GS), achieving a $\sim 50\%$ reduction in synchronization errors and a PSNR improvement of up to **4dB**, thereby outperforming existing state-of-the-art methods.

## 1 Introduction

Novel view synthesis, which takes a set of 2D images from different views and outputs realistic images from arbitrary views, is a classical task in both the communities of computer vision and computer graphics, serving as the basis for creating immersive contents for virtual/augmented/mixed reality. This topic is drawing booming attentions due to the inventions of neural radiance fields (NeRF) (Mildenhall et al., 2021) and Gaussian splatting (Kerbl et al., 2023), massive papers have been published and the research scope has been extended from modeling the static world to dynamic one with multi-view videos input.

However, all of the existing techniques (Li et al., 2022; Park et al., 2021a; Shao et al., 2023; Fridovich-Keil et al., 2023; Cao & Johnson, 2023; Pumarola et al., 2021; Gao et al., 2024; Wu et al., 2024; Yang et al., 2024; Xu et al., 2024; Luiten et al., 2023; Li et al., 2024; Katsumata et al., 2024; Fang et al., 2022) rely on accurate synchronization between videos captured from different cameras. To achieve synchronization between cameras, researchers typically rely on hardware synchronization via dedicated sync cables connecting the devices. This not only increases the cost of the system but also compounds its complexity, thereby reducing the system's portability and usability and limiting the application and promotion of NeRF and Gaussian splatting techniques. Although a few of papers (Kim et al., 2024; Choi et al., 2024) have been published to synchronize without relying on hardware-based solutions and setting temporal offsets as learnable variables within dynamic reconstruction framework, these methods could only process videos with small temporal misalignment and rely heavily on scenes with significant textures, or restricted to human-centric scenes.

The key of achieving synchronization according to multi-view videos only is finding unique priors which identify consistency not only across different views but also along time stamps. However, the most commonly used photometric consistency prior is prone to local minima optimization when processing low/repeated-texture areas, where gradients are weak and provide little guidance, and multiple candidate correspondences produce ambiguous matches that can mislead the solver and slow convergence. This ill-posedness is further deteriorated when extending the static novel view

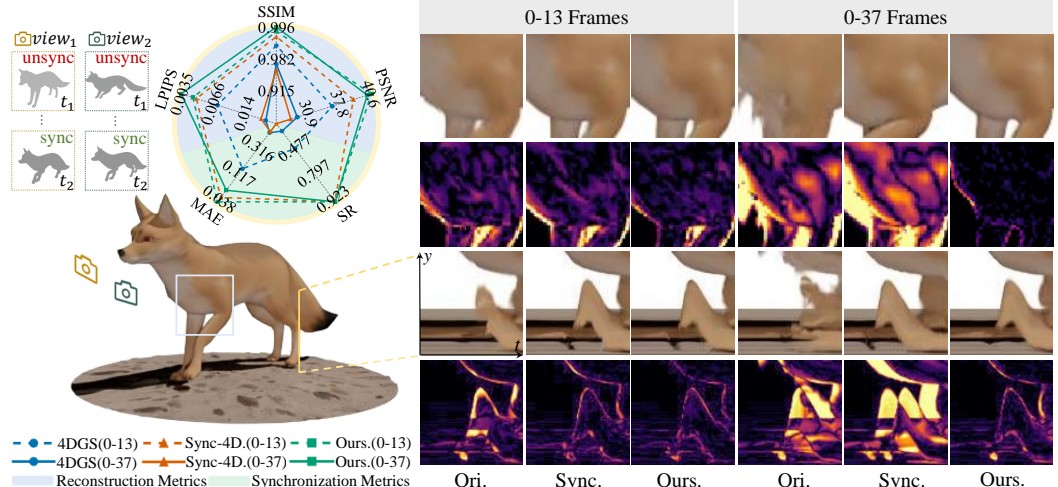

Figure 1: Our method significantly enhances reconstruction quality and temporal consistency compared to baseline methods across different unsynchronized offset settings. **Note:** 0–13 and 0–37 indicate the ranges of absolute unsynchronized offsets in frames. **Left**: The radar chart shows our quantitative advantages. **Right**: The novel view results, $y$–$t$ slices, and error maps demonstrate better temporal consistency and reduced ghosting within our approach.

synthesis to dynamic one, where the low/repeated textures in spatial domain are evolved in the one along both the spatial and temporal domains, resulting in unreliable optimization of temporal offset.

To address the unreliable synchronization optimization, we propose the global-local motion consistency prior. We notice that the local 2D optical flows projected from the global 3D scene flow differ significantly from each other due to the camera poses, contrary to the photometric consistency where the projected appearances in different cameras changes slowly. As a result, the motion consistency is inherited more suitable for identifying the temporal misalignment than the appearance. Following this prior, we propose the global-local motion consistency loss function by comparing the projected optical flow with the pre-calculated one. To verify the efficacy of the proposed global-local motion consistency loss function, we integrate it with the most popular dynamic novel view synthesis frameworks, including the dynamic NeRF (Fridovich-Keil et al., 2023) and Gaussian Splatting (Wu et al., 2024; Katsumata et al., 2024). Experiments on public datasets (Li et al., 2022; Kim et al., 2024; Abou-Chakra et al., 2024) with various synchronization settings reveal that the proposed loss function could provide consistent performance for unsynchronized videos with both small and large temporal misalignment, and not only reduces the synchronization error by a half, but also improves the quality of novel view synthesis by $\sim 4$ **dB**. Specifically, we make the following contributions,

1. We demonstrate that motion features are more reliable than photometric cues for temporal offset optimization of unsynchronized multi-view videos.
2. We identify the global-local motion consistency prior and integrate it with popular frameworks, *i.e.*, dynamic NeRF and Gaussian Splatting, to improve novel view synthesis under unsynchronized conditions.
3. We substantiate that the proposed global-local motion consistency loss surpasses prior works, reducing synchronization errors by a half and improving the PSNR by $\sim 4$ dB.

## 2  RELATED WORKS

### 2.1  DYNAMIC NOVEL VIEW SYNTHESIS

Dynamic novel view synthesis, which models spatiotemporal variations of object motion and appearance, has advanced through neural scene representations. Building on NeRF's (Mildenhall et al., 2021) static formulation, subsequent works (Zhu et al., 2023; Liu et al., 2024; Zhu et al., 2024) advanced INR modeling and optimization, and progress in light-field rendering (e.g., Geo-NI (Wu et al., 2025)) explores geometry-aware neural interpolation, while a parallel line of temporal exten-

sions such as Fang et al. (2022); Park et al. (2021b); Xu et al. (2024); Fridovich-Keil et al. (2022); Li et al. (2022); Shao et al. (2023); Park et al. (2021a); Fridovich-Keil et al. (2023); Cao & Johnson (2023); Pumarola et al. (2021) address dynamic scenes through diverse strategies. Pioneering works like D-NeRF (Pumarola et al., 2021) established canonical space mapping through time-varying deformation fields for photorealistic synthesis. NSFF (Li et al., 2021b) introduces neural scene flow fields to jointly model geometry and 3D motion, enforcing 3D–2D flow-based consistency for dynamic view synthesis. Subsequent advancements such as K-Planes (Fridovich-Keil et al., 2023) and HexPlanes (Cao & Johnson, 2023) further enhanced reconstruction efficiency through tensor decomposition-based multi-plane factorization. Notably, the emergence of 3DGS (Kerbl et al., 2023) marked a paradigm shift through anisotropic Gaussian primitives and CUDA-accelerated rasterization, achieving real-time rendering without compromising quality and overcoming the limitations of implicit representations. With the advent of 3DGS, some methods (Wu et al., 2024; Yang et al., 2023; 2024; Luiten et al., 2023; Li et al., 2024; Katsumata et al., 2024) have extended 3DGS to the dynamic domain. Yang et al. (2024) employ canonical space deformation fields to model Gaussian transformations. 4DGS (Wu et al., 2024) innovates through neural voxel encoding and lightweight MLP-based deformation prediction. EDGS (Katsumata et al., 2024) introduces a compact dynamic 3D Gaussian representation with time-varying Gaussian parameters equipped with basis functions for representing dynamic scenes. Recent SC-GS (Li et al., 2024) further enhances temporal modeling with dedicated motion parameters and time-dependent opacity control.

Despite these advancements, current dynamic novel view systhesis methods share a critical limitation: they strictly require synchronized multi-view inputs. Unsynchronized video sequences often cause reconstruction failures, particularly in fast-moving regions, due to the lack of a globally aligned timeline. Although a few methods (Kim et al., 2024; Choi et al., 2024) have been proposed, their applicability is limited. Sync-NeRF (Kim et al., 2024) sets learnable per-camera temporal offsets, but is restricted to textured scenes with only small temporal misalignments. Choi et al. (2024) introduces human pose priors, but is restricted to human-centric scenes, whereas our approach is applicable to general dynamic scenes.

### 2.2 OPTICAL FLOW FOR SYNCHRONIZATION

Motion features have been utilized for synchronizing multi-videos for a long time (Wolf & Zomet, 2006; Pundik & Moses, 2010). Recently, Huo et al. (2020) propose a reference frame alignment method for frame extrapolation to establish nonlinear temporal correspondence between videos. Purushwalkam et al. (2020) propose an alignment procedure to connect patches between videos via cross-video cycle consistency. Wu et al. (2019) propose a new deep network called SynNet to synchronize multiple motion-camera videos by exploiting and matching view invariant pose features. Optical flow (Hur & Roth, 2019; Teed & Deng, 2020; Li et al., 2021a), as one of the most widely used representations of motion, frequently appears in various synchronization tasks. Cam-LiFLOW (Liu et al., 2022) takes two consecutive synchronized camera and Lidar frames as inputs to estimate the optical flow and scene flow simultaneously and builds multiple bidirectional connections between its 2D and 3D branches. ROFT (Piga et al., 2021) exploits real-time optical flow to synchronize delayed instance segmentation and 6D object pose estimation streams. These prior works demonstrate the strong effectiveness of motion cues but are limited to settings with significant view overlap and do not address joint reconstruction for multi-view inputs with minimal shared content and temporal misalignment.

## 3 METHOD

### 3.1 PRELIMINARY

In dynamic scene reconstruction, temporal alignment serves as a fundamental requirement for establishing consistent spatiotemporal representations across asynchronous multi-view observations. We represent the dynamic scene as a continuous function $\mathbf{F}_t(\vec{x}, \vec{d})$ that encodes spatial coordinates $\vec{x} \in \mathbb{R}^3$, viewing direction $\vec{d} \in \mathbb{S}^2$, and temporal dimension $t \in \mathbb{R}^+$. This neural representation can be formulated as,

$$\mathbf{F}_t = \mathcal{R}\left(\{\mathcal{P}_i, I_i(t)\}_{i=1}^M\right),\tag{1}$$

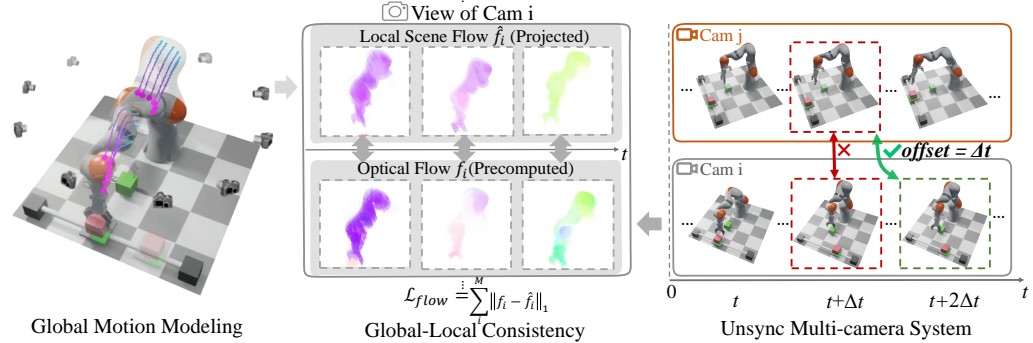

Figure 2: Illustration of the global-local motion consistency framework. Our framework introduces Global-Local Motion Consitency that jointly optimizes 3D scene flow estimation and dynamic scene reconstruction. The pipeline operates on two complementary levels: (1) Local 2D optical flows extracted from multi-view video streams, and (2) Global 3D scene flows derived via projection of neural radiance fields. Our method establishes geometric consistency between image-space flow warping and 3D scene flow reprojection through co-optimization of global-local constraints, enabling simultaneous spatiotemporal alignment and geometry reconstruction within an end-to-end trainable architecture.

where $\mathcal{R}(\cdot)$ denotes the reconstruction operator, which can be implemented using methods like NeRF or 3D Gaussian Splatting to synthesize the scene from M camera observations. Each camera provides a video stream $\{I_i(t)\}_{t=1}^{N}$ (with N temporal samples) along with projection matrixes $\mathcal{P}_i = \{\mathbf{K}_i, \mathbf{E}_i\}$, where $\mathbf{K}_i$ and $\mathbf{E}_i$ represent intrinsic and extrinsic matrices, respectively.

To address temporal misalignment between asynchronous cameras, temporal offsets per camera $\{\Delta t_i\}_{i=1}^{M}$ are introduced to compensate for unsynchronized frame captures. We focus on correcting large frame-level temporal misalignments and do not account for minor sub-frame variations induced by other camera effects(e.g., rolling shutter, clock drift, and exposure differences). Finally, the scene and the temporal offsets are optimized by minimizing the RGB loss function, *i.e.*,

$$
\begin{aligned}
\mathbf{F}_t^*, \{\Delta t_i^*\}_{i=1}^{M} &= \underset{\mathbf{F}_t, \{\Delta t_i\}_{i=1}^{M}}{\arg\min} \ \mathcal{L}_{\text{RGB}} \\
&= \underset{\mathbf{F}_t, \{\Delta t_i\}_{i=1}^{M}}{\arg\min} \ \sum_{i=1}^{M} \|\pi(\mathbf{F}_t, \mathcal{P}_i) - \mathbf{I}_i(t + \Delta t_i)\|_2^2 ,
\end{aligned}
\tag{2}
$$

where $\pi(\cdot)$ denotes the projection operator that projects the scene representation onto the image plane using the camera projection matrix $\mathcal{P}_i$. This formulation enables joint optimization of scene geometry, appearance, and temporal offsets through gradient-based methods.

## 3.2 MOTION CONSISTENCY ENHANCED UNSYNC. D-NVS

### 3.2.1 MOTION CONSISTENCY PRIOR

Despite the fact that the exploration mentioned in preliminary has achieved some success in certain scenarios, the inborn ambiguity of the photometric consistency results in an unreliable optimization of the temporal offsets and the reconstructed dynamic scenes especially when the scenes are composed of weakly textured or textureless regions. As shown in the Fig. 3, an actress dressed entirely in blue is moving with the rhythm of time. Two points (marked as red and orange, respectively) are tracked across different frames (left-bottom panel). Because of the isotropy of the pure blur color, it is almost impossible to build correct correspondences between different frames and views for these two

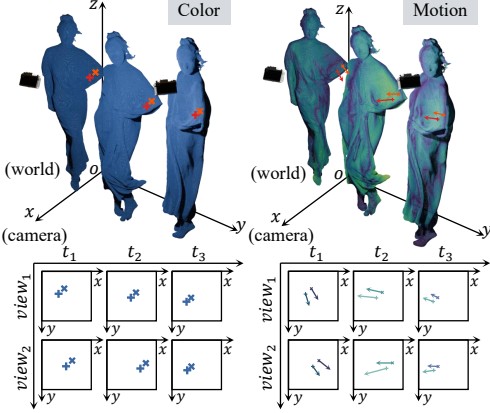

Figure 3: Comparison between color and motion clues for D-NVS across temporal-view dimensions.

points. As a result, existing dynamic novel view synthesis frameworks with only color supervision (*i.e.*, the Eqn. 2) fail in optimizing $\mathbf{F}_t^*$ and $\{\Delta t_i^*\}_{i=1}^M$.

On the contrary, the inborn ambiguity of photometric consistency-based supervision could be fundamentally eliminated by introducing the motion consistency. As shown in the right half of the Fig. 3, the marked two points shared similar 3D scene flow vectors. Due to the perspective transformation caused by camera's projection matrixes, similar 3D scene flow vectors will be projected to 2D optical flows which differ significantly from each other (*i.e.*, the amplitude and the direction), as shown in the right-bottom panel of the Fig. 3. Leveraged from these unique view/time-dependent characteristics, both the optimizations for $\mathbf{F}_t^*$ and $\{\Delta t_i^*\}_{i=1}^M$ could be improved by introducing the motion consistency prior into the Eqn. 2, *i.e.*, the projected scene flow should be equal to the inherit optical flow in each input video.

### 3.2.2 GLOBAL-LOCAL MOTION CONSISTENCY FOR D-NVS

Following the motion consistency previously analyzed, we propose the global-local motion consistency loss function by comparing the projected scene flow with the inherit optical flow, *i.e.*,

$$\mathcal{L}_{\text{Flow}} = \sum_{i=1}^{M} \left\| f_i(t) - \hat{f}_i(t + \Delta t_i) \right\|_1 , \tag{3}$$

where $\hat{f}_i$ and $f_i$ refer to the projected scene flow according to projection matrix $\mathcal{P}_i$ of the $i$-th camera and the optical flow in the video stream provided by the $i$-th camera, respectively. $\| \cdot \|$ calculates the $L_1$-norm of vector $\cdot$.

The projected scene flow $\hat{f}_i$ could be obtained by applying the 3D flow operator $\mathcal{F}_{3D}$ to the scene function $\mathbf{F}$, and then applying the project operator $\pi$ according to the projection matrix,

$$\hat{f}_i = \pi(\mathcal{F}_{3D}(\mathbf{F}_t), \mathcal{P}_i). \tag{4}$$

Note that, we only provide a general form here. The 3D flow operator $\mathcal{F}_{3D}$ varies from each other, and is determined by the adopted NeRF, GS or other frameworks, we provide detailed discussions in the experimental section.

The optical flow $f_i$ could be pre-calculated using the operator $\mathcal{F}_{2D}$,

$$f_i = \mathcal{F}_{2D}(I_i(t)). \tag{5}$$

Note that, we do not restrict the operator $\mathcal{F}_{2D}$ to be a specific optical flow algorithm. Actually, as shown in the ablation study section, both the algorithms published in 2018 and 2023 could provide significantly improvements compared with previous methods. Fig. 2 visualizes the pipeline of the global-local motion consistency for unsynchronized dynamic novel view synthesis.

However, as analyzed in many optical flow papers, it is difficult to provide reliable predictions for low/repeated-texture regions. As a result, the reconstruction accuracy of $\mathbf{F}_t^*$ and $\{\Delta t_i^*\}_{i=1}^M$ will be reduced if unreliable flow is used. Fortunately, we notice that *All pixels in a frame from any camera share a same temporal offset.* Following this observation, we introduce a reliability masking strategy to filter out unreliable pixels, and thus the Eqn. 3 is modified as

$$\mathcal{L}_{\text{Flow}} = \sum_{i=1}^{M} \left\| \mathbf{m}_i \odot \left( f_i(t) - \hat{f}_i(t + \Delta t_i) \right) \right\|_1 , \tag{6}$$

where $\odot$ denotes element-wise multiplication and $\mathbf{m}_i$ denotes a binary reliability mask, which is implemented by selecting the pixels which have the largest 50% optical flow amplitudes.

The final loss function integrates multiple constraints, including the RGB loss, the proposed motion consistency loss, and a regularization term on $\Delta t$ to prevent excessive time shifts, *i.e.*,

$$\mathbf{F}_t^*, \{\Delta t_i^*\}_{i=1}^M = \underset{\mathbf{F}_t, \{\Delta t_i\}_{i=1}^M}{\arg\min} \; \mathcal{L}_{\text{RGB}} + \lambda_{\text{Flow}} \, \mathcal{L}_{\text{Flow}} + \lambda_t \sum_{i=1}^{M} |\Delta t_i| , \tag{7}$$

where $\lambda_{\text{Flow}}$ and $\lambda_t$ control the relative weights of the flow consistency and temporal regularization, respectively. To ensure consistency with Sync-NeRF, we include the temporal regularization term $\lambda_t$, but set it to a negligible weight to avoid overly strong constraints. By jointly optimizing these terms, we achieve robust, temporally aligned, and motion-consistent dynamic scene reconstruction.

Table 1: Metrics comparing the original framework, offset-optimized framework, and our global-local motion prior-optimized framework on unsynchronized Plenoptic and Blender datasets across varying camera offsets, with the Kplanes baselines excluded from ParticleNeRF's scenes.

| | | | 4DGS | | | EDGS | | | Kplanes | | | Improvement | |
|---|---|---|---|---|---|---|---|---|---|---|---|---|---|
| Offset | Metric | Ori. | Sync. | Ours | Ori. | Sync. | Ours | Ori. | Sync. | Ours | vs Ori. | vs Sync. |
| **Plenoptic Datasets** 0-13 Frames | PSNR↑ | 30.27 | 30.84 | **31.29** | 28.07 | 27.62 | **29.72** | 29.65 | 30.12 | **30.45** | **+1.16** | **+0.96** |
| | SSIM↑ | 0.9376 | 0.9392 | **0.9403** | 0.9280 | 0.9245 | **0.9344** | 0.9188 | 0.9251 | **0.9264** | **+0.0056** | **+0.0041** |
| | LPIPS↓ | 0.1003 | 0.0979 | **0.0970** | 0.1052 | 0.1025 | **0.0995** | 0.2103 | 0.1997 | **0.1992** | **-0.0067** | **-0.0015** |
| 0-21 Frames | PSNR↑ | 29.50 | 30.16 | **30.59** | 28.26 | 27.80 | **29.34** | 28.42 | 28.62 | **29.45** | **+1.07** | **+0.93** |
| | SSIM↑ | 0.9280 | 0.9321 | **0.9336** | 0.9231 | 0.9195 | **0.9297** | 0.9062 | 0.9090 | **0.9164** | **+0.0075** | **+0.0064** |
| | LPIPS↓ | 0.1245 | 0.1173 | **0.1156** | 0.1246 | 0.1250 | **0.1168** | 0.2255 | 0.2206 | **0.2069** | **-0.0118** | **-0.0079** |
| 0-33 Frames | PSNR↑ | 28.79 | 29.80 | **30.65** | 27.88 | 28.03 | **30.05** | 28.44 | 27.88 | **29.10** | **+1.56** | **+1.36** |
| | SSIM↑ | 0.9256 | 0.9292 | **0.9340** | 0.9210 | 0.9212 | **0.9331** | 0.9003 | 0.9003 | **0.9120** | **+0.0107** | **+0.0095** |
| | LPIPS↓ | 0.1246 | 0.1171 | **0.1159** | 0.1253 | 0.1200 | **0.1172** | 0.2297 | 0.2295 | **0.2131** | **-0.0111** | **-0.0068** |
| **Blender Datasets** 0-13 Frames | PSNR↑ | 31.61 | 33.75 | **36.46** | 32.24 | 35.52 | **36.68** | 32.92 | 39.11 | **39.14** | **+5.17** | **+1.30** |
| | SSIM↑ | 0.9711 | 0.9815 | **0.9853** | 0.9704 | 0.9855 | **0.9873** | 0.9764 | **0.9855** | 0.9848 | **+0.0132** | **+0.0016** |
| | LPIPS↓ | 0.0265 | 0.0135 | **0.0121** | 0.0276 | 0.0105 | **0.0095** | 0.0297 | **0.0158** | 0.0163 | **-0.0153** | **-0.0006** |
| 0-23 Frames | PSNR↑ | 26.51 | 28.06 | **33.50** | 28.03 | 28.50 | **31.37** | 29.82 | 29.54 | **30.59** | **+3.70** | **+3.12** |
| | SSIM↑ | 0.9568 | 0.9637 | **0.9751** | 0.9593 | 0.9604 | **0.9738** | 0.9695 | 0.9700 | **0.9725** | **+0.0119** | **+0.0091** |
| | LPIPS↓ | 0.0373 | 0.0213 | **0.0127** | 0.0381 | 0.0232 | **0.0229** | 0.0322 | 0.0270 | **0.0221** | **-0.0166** | **-0.0046** |
| 0-37 Frames | PSNR↑ | 24.72 | 26.05 | **30.21** | 26.30 | 29.67 | **32.01** | 28.47 | 28.72 | **29.08** | **+3.94** | **+2.29** |
| | SSIM↑ | 0.9495 | 0.9558 | **0.9640** | 0.9507 | 0.9611 | **0.9662** | 0.9651 | 0.9669 | **0.9690** | **+0.0113** | **+0.0051** |
| | LPIPS↓ | 0.0421 | 0.0305 | **0.0249** | 0.0414 | 0.0266 | **0.0263** | 0.0386 | 0.0351 | **0.0274** | **-0.0145** | **-0.0045** |

## 4 EXPERIMENTS

### 4.1 EXPERIMENTAL SETTINGS

**Unsynchronized datasets and metrics.** We evaluated our method on an unsynchronized Plenoptic Dataset (derived from the Plenoptic Video Dataset (Li et al., 2022) with 6 scenes) and an unsynchronized Dynamic Blender Dataset with 3 scenes, both provided by Sync-NeRF (Kim et al., 2024), as well as two scenes from the Blender Datasets provided by ParticleNeRF (Abou-Chakra et al., 2024). To test the robustness of our method under larger synchronization challenges, we further extend the datasets by introducing random integer temporal offsets in both synthetic and real datasets. During training, the model is allowed to learn continuous (non-integer) offsets. For the Blender Datasets, the absolute offset ranges are set to 0–23 and 0–37 frames, and for the Plenoptic Dataset, 0–21 and 0–33 frames, where the actual offsets are randomly sampled within these ranges (including negative values). This results in average misalignments of approximately 10–11 and 14–15 frames, respectively.

To evaluate the quality of the rendered images, we use several commonly used metrics: PSNR, SSIM, and LPIPS. For the unsynchronized Blender Datasets with ground truth, we additionally calculate MAE (Mean Absolute Error) and SR (Success Rate, defined as the percentage of predictions with offset errors ≤ a pre-defined threshold, e.g., 3 frames) to assess the accuracy of the predicted time offsets.

**Baselines.** To validate the universality of our approach across different scene representations, we adopt our method on the original baselines: 4DGS (Wu et al., 2024), EDGS (Katsumata et al., 2024) and Kplanes (Fridovich-Keil et al., 2023) with hybrid encoder. Similarly, we applied the Sync-NeRF method to these baselines, named Sync-4DGS, Sync-EDGS, and Sync-Kplanes.

Table 2: Performance comparison of temporal metrics on Blender Datasets with Kplanes baselines excluded from ParticleNeRF's scenes. Metrics (unit): MAE: Mean absolute error (second), SR: Success rate (%), defined as the percentage of predictions with offset errors within a pre-defined threshold. To avoid bias from a single threshold, we report Avg.SR, the average success rate over thresholds from 3 to 10 frames.

| | | 4DGS | | EDGS | | Kplanes | |
|---|---|---|---|---|---|---|---|
| Offset | Metric | Sync. | **Ours** | Sync. | **Ours** | Sync. | **Ours** |
| 0-13 Frames | MAE↓ | 0.062 | **0.029** | 0.047 | **0.026** | **0.015** | **0.015** |
| | Avg.SR↑ | 94.6 | **99.7** | 96.2 | **100.0** | **100** | **100** |
| 0-23 Frames | MAE↓ | 0.163 | **0.083** | 0.207 | **0.139** | 0.319 | **0.176** |
| | Avg.SR↑ | 70.0 | **91.9** | 47.9 | **75.5** | 24.4 | **62.2** |
| 0-37 Frames | MAE↓ | 0.319 | **0.141** | 0.262 | **0.171** | 0.460 | **0.315** |
| | Avg.SR↑ | 28.5 | **70.0** | 34.3 | **60.5** | 20.5 | **31.1** |

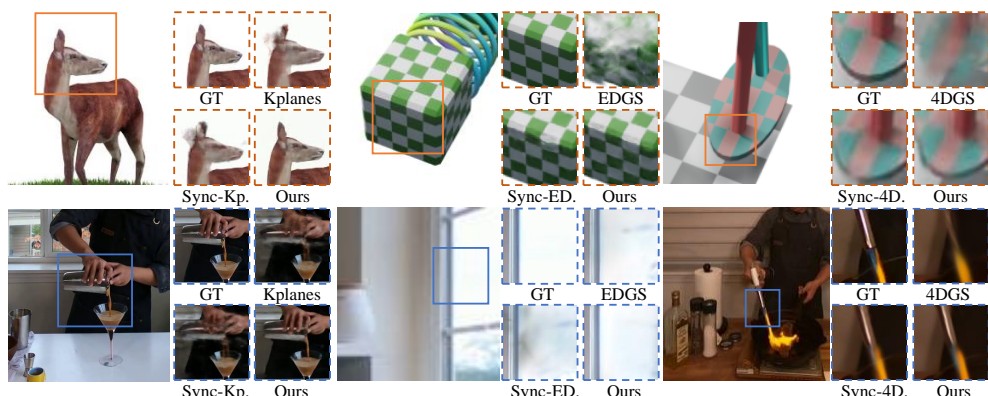

Figure 4: Compared to other methods, our approach better preserves structural details and fine textures. Notably, our method retains sharper contours, clearer boundaries, and smoother surface details in challenging dynamic scenes.

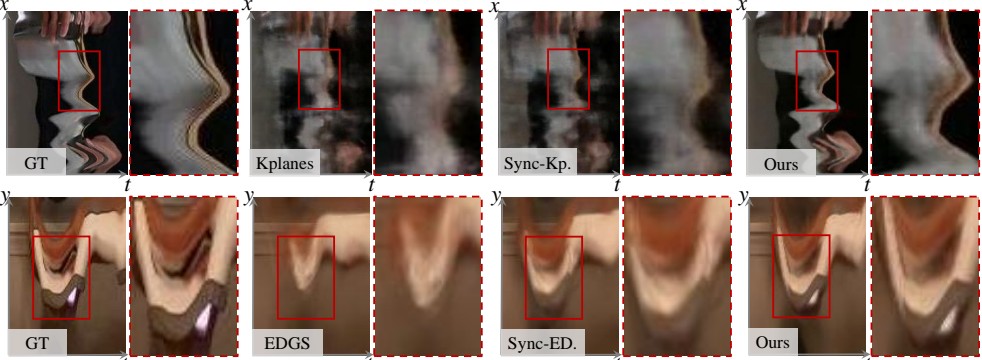

Figure 5: Qualitative comparison of spatiotemporal image rendering. Our method demonstrates enhanced detail preservation and spatiotemporal coherence in temporal-axis sampling slices, with improved dynamic range compared to baseline approaches.

**Explicit Gaussian Representations.** To cover the mainstream approaches of dynamic Gaussian modeling, we adopt 4DGS and EDGS as representative frameworks, which capture temporal dynamics by using a deformation field and by directly parameterizing Gaussian centers as time-dependent functions, respectively. For EDGS (Katsumata et al., 2024), the implementation of the 3D flow operator $\mathcal{F}_{3D}$ is carried out by tracking time-parameterized 3D Gaussians' motions between consecutive time steps, and the projected scene flow $\hat{f}_i$ is obtained via projection $\pi(\cdot)$, blending contributions with Gaussian weights. For 4DGS (Wu et al., 2024), inspired by Gao et al. (2024), the 3D flow operator $\mathcal{F}_{3D}$ is implicitly realized through splatting. The projected scene flow $\hat{f}_i$ is then obtained by measuring the displacement of each Gaussian's 2D mean position between consecutive frames, weighted by its contribution to the pixel.

**Hybrid Factorization Representations.** Hybrid methods (e.g., KPlanes (Fridovich-Keil et al., 2023)) balance efficiency-accuracy trade-offs via combined explicit-implicit representations. For Kplanes, the 3D flow operator $\mathcal{F}_{3D}$ is implemented by first extracting features from six orthogonal planes and feeding them into a scene flow decoder to predict per-point 3D motion. The projected scene flow $\hat{f}_i$ is then obtained via the projection operator $\pi(\cdot)$, by transforming the predicted 3D flow into the image plane using the camera's intrinsic and extrinsic parameters.

## 4.2 RESULTS

### 4.2.1 SPATIAL CONSISTENCY.

Across both Plenoptic and Blender Datasets under varying temporal offsets, our method consistently improves spatial coherence and novel view synthesis quality. *Under larger offset ranges, absolute performance may fluctuate across datasets due to differences in their frame ranges and sequence*

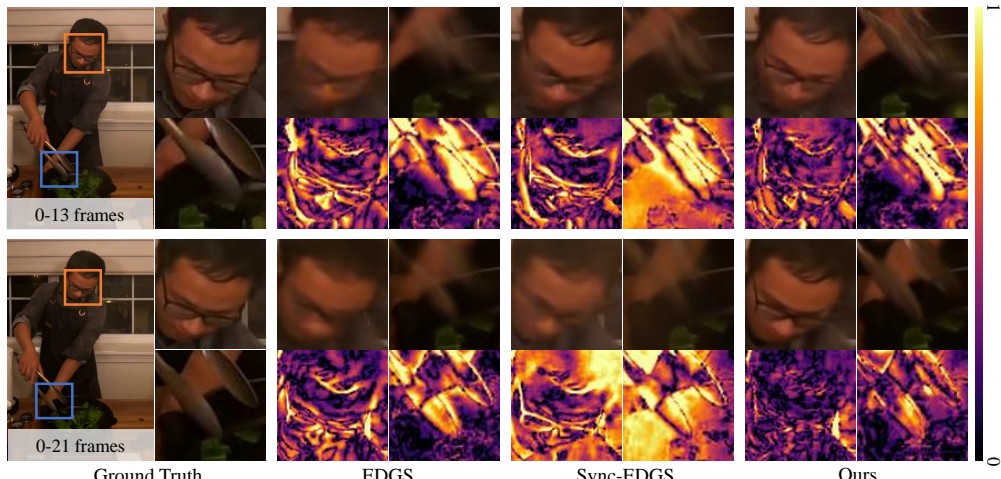

Figure 6: Comparison of error map under different temporal offsets. Our method exhibits more pronounced performance improvements under larger temporal offsets, particularly in dynamic textureless regions where motion artifacts are predominant.

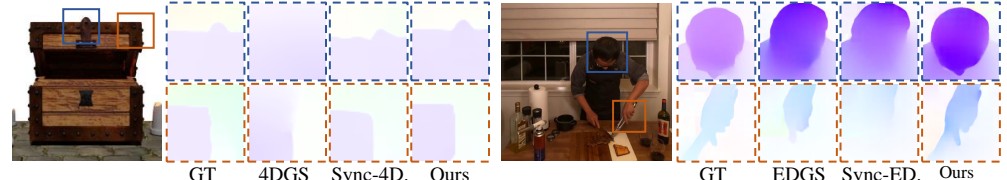

Figure 7: Optical flow comparison: Our method matches GT flow in key regions (color/contour accuracy) vs baselines.

*lengths.* As shown in Table 1, our method outperforms all baselines, achieving average gains of at least 1.30 dB and 0.93 dB on the Blender and Plenoptic Datasets respectively. Notably, performance remains stable even with increasing misalignment, demonstrating strong adaptability to unsynchronized inputs. The qualitative results in Figure 4 show that the baseline methods suffer from structural artifacts, such as unclear textures, object boundary distortion, and ghosting, for instance, the outline of a coffee cup and the ears of a deer. In contrast, our method preserves object silhouettes and geometric integrity, producing spatially consistent, visually coherent novel views. Error maps in Figure 1 further highlight improvements in regions like the texture of fox fur. Additionally, flow maps in Figure 7 show that our synthesized views align more closely with ground-truth motion, reducing ambiguities in challenging regions like finger joints and box edges.

### 4.2.2 TEMPORAL CONSISTENCY.

Our method also achieves strong temporal consistency across dynamic scenes. As shown in Figure 5, temporal-axis slices demonstrate that our approach preserves coherent motion over time, while Sync-Kplanes and Sync-EDGS suffer from noticeable blurring and texture inconsistency in terms of both color and contour, especially in regions with fine structures and significant motion such as handheld tools and the reflective surfaces of cups. In contrast, our method reconstructs the spring's subtle oscillations with smooth temporal transitions, avoiding motion blur and edge instability. Figure 1 further confirms this: as temporal misalignment increases from 0–13 to 0–37 frames, baseline methods exhibit significant error accumulation. In contrast, our results remain stable, with error maps consistently aligned with ground truth. This robustness against temporal misalignment highlights the effectiveness of our global-local motion consistency prior and ensuring reliable performance despite increasing synchronization challenges.

As shown in Table 2, our method achieves the highest MAE reduction of **56%** (0.141s vs 0.319s at 0-37 frames) and the highest SR improvement of **2.6**× (62.2% vs 24.4% at 0-23 frames) among all evaluated models and offset ranges, demonstrating superior robustness to severe temporal misalignment.

## 4.3 ABLATION STUDY

**Effectiveness of Flow Supervision and Reliability Masking.** To evaluate the contributions of the flow-consistency supervision and the reliability masking strategy, we conduct an ablation study on the Plenoptic Datasets on the EDGS baseline. In the ablated variants, either the flow term is removed (w/o flow) or the reliability mask is removed (w/o Reliability Mask). As shown in Table 3, removing the flow term leads to a large performance drop across all offset ranges, demonstrating that global-local flow consistency is the dominant contributor to synchronization quality. Removing the reliability mask while keeping the flow term results in a smaller but noticeable drop in performance, indicating that the mask further improves optimization by focusing the loss on dynamically reliable regions. Overall, our full method with both flow supervision and reliability masking achieves the best reconstruction quality.

Table 3: Quantitative ablation on the reliability mask and flow supervision under different offset ranges.

| Offset | Model | PSNR | SSIM | LPIPS |
|---|---|---|---|---|
| 0-13 Frames | Ours | **29.72** | 0.9344 | **0.0995** |
| | w/o Reliability Mask | 29.53 | **0.9850** | 0.1021 |
| | w/o flow | 27.62 | 0.9245 | 0.1025 |
| 0-21 Frames | Ours | **29.34** | **0.9297** | **0.1168** |
| | w/o Reliability Mask | 28.77 | 0.9260 | 0.1216 |
| | w/o flow | 27.80 | 0.9195 | 0.1168 |
| 0-33 Frames | Ours | **30.05** | **0.9331** | 0.1172 |
| | w/o Reliability Mask | 29.38 | 0.9294 | **0.1153** |
| | w/o flow | 28.03 | 0.9212 | 0.1200 |

**Robustness to Flow Estimation Qualities.** To further validate the effectiveness of our proposed reliability masking strategy, we conducted an ablation study on the 4DGS baseline using PWC-Net (Sun et al., 2018), a classical optical flow algorithm, to generate the precomputed flow on the Blender Datasets provided by Sync-NeRF. Although PWC-Net is significantly less accurate than recent models such as VideoFlow (Shi et al., 2023) (used in our 4DGS baseline), the results in Table 4 show that with our masking strategy, the reconstruction performance not only surpasses the Sync., but also approaches the quality achieved with high-precision flow estimation. These results highlight that our mask is robust to flow quality and can effectively suppress unreliable supervision.

Table 4: Ablation with PWC-Net flow surpasses Sync. and nearly matches VideoFlow performance.

| Offset | Model | PSNR | SSIM | LPIPS |
|---|---|---|---|---|
| 0-13 Frames | Sync. | 35.02 | 0.9810 | 0.0172 |
| | Ours(PWC-Net) | 36.96 | 0.9836 | **0.0151** |
| | Ours(VideoFlow) | **37.60** | **0.9843** | **0.0151** |
| 0-23 Frames | Sync. | 29.95 | 0.8727 | 0.0195 |
| | Ours(PWC-Net) | 35.80 | 0.9816 | 0.0150 |
| | Ours(VideoFlow) | **36.24** | **0.9818** | **0.0076** |
| 0-37 Frames | Sync. | 27.08 | 0.9652 | 0.0253 |
| | Ours(PWC-Net) | 32.05 | 0.9766 | 0.0181 |
| | Ours(VideoFlow) | **34.74** | **0.9802** | **0.0162** |

**The weight of the flow Term.** For 4DGS, $\lambda_{\text{Flow}}$ is set to 0.05 for real-world scenes and 0.5 for blender scenes. To study the effect of varying the weight of the flow consistency loss, $\lambda_{\text{Flow}}$, we conduct an ablation study on the 0-13 frames unsynchronized Plenoptic Dataset using the 4DGS baseline. The results are shown in Table 5. These results demonstrate that an appropriate choice of the flow-loss weight, $\lambda_{\text{Flow}}$, is essential for the effectiveness of our method. When $\lambda_{\text{Flow}}$ is set too low, the influence of the flow-supervision term is weakened, which reduces its ability to enforce temporal consistency and leads to a noticeable degradation in reconstruction quality.

Table 5: Ablation on the flow-loss weight $\lambda_{\text{Flow}}$.

| Metric | $\lambda_{\text{Flow}} = 0.1$ | $\lambda_{\text{Flow}} = 0.05$ | $\lambda_{\text{Flow}} = 0.01$ | $\lambda_{\text{Flow}} = 0.001$ |
|---|---|---|---|---|
| PSNR | 31.05 | **31.29** | 30.94 | 30.81 |
| SSIM | 0.9397 | **0.9403** | 0.9402 | 0.9380 |
| LPIPS | 0.0979 | **0.0970** | 0.0974 | 0.0987 |

## 5 CONCLUSION AND FUTURE WORK

In this work, we propose a novel framework for dynamic scene reconstruction that effectively addresses temporal misalignment in unsynchronized multi-view videos. By leveraging global-local motion consistency, our method achieves robust temporal alignment and high-fidelity reconstruction. It applies seamlessly to both explicit Gaussian representations and implicit neural radiance fields, offering a unified solution for diverse scene types. Experiments on real-world and synthetic datasets demonstrate the superior performance of our method in handling large temporal misalignments, reducing motion artifacts while preserving fine-grained details.

Although our framework achieves strong performance across diverse settings, it still relies on a relatively simple reliability mask. While effective in most cases, it may struggle under heavy occlusion. Incorporating more advanced visibility checks is a promising direction to enhance robustness. In ad-

dition, exploring hierarchical spatio-temporal pyramids may improve motion supervision for videos with very large frame-level misalignments. Moreover, more complex camera effects such as rolling shutter, clock drift, and exposure differences are also worth considering, and addressing them will require both methodological extensions and richer dataset support, which we leave for future work.

ACKNOWLEDGMENTS

We solely utilized LLMs for language polishing and grammar correction. LLMs were not involved in the derivation of scientific content and conclusions.

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

# A    APPENDIX

**Multi-scale Flow supervision.** In the quantitative analysis of our paper, we employed a single-scale loss function. We also explored the potential advantages of a multi-scale loss approach. Our study on multi-scale flow supervision using Plenoptic Datasets with the 4DGS baseline revealed that supplying motion information at various resolutions enables the model to efficiently capture both extensive global motions and fine local details. This multi-scale method bolsters the model's robustness against changes in motion scale and helps to avoid overfitting to a single scale, as demonstrated by its improved performance over the single-scale counterpart in Table 6.

Table 6: Multi-scale flow metrics show consistent gains over single-scale supervision.

| Model | PSNR | SSIM | LPIPS |
|---|---|---|---|
| Ours(0-13) | **31.29** | 0.9403 | **0.0970** |
| Ours-multi(0-13) | 31.26 | **0.9409** | **0.0970** |
| Ours(0-21) | 30.59 | 0.9336 | 0.1156 |
| Ours-multi(0-21) | **30.95** | **0.9354** | **0.1151** |
| Ours(0-33) | 30.65 | **0.9340** | 0.1159 |
| Ours-multi(0-33) | **30.82** | **0.9340** | **0.1150** |

**Additional Quantitative Results on Synchronized Videos.** To further validate the effectiveness of our method, we report additional results on synchronized videos. Table 7 presents the average PSNR performance across different time offset ranges, evaluated against the ground-truth (GT) offset videos.

Table 7: Quantitative results of synchronized videos. Average PSNR across different time offset ranges, compared against GT-offset results(videos pre-aligned using ground-truth temporal offsets).

| Baseline | Dataset | Average PSNR | | | |
| | | Ori. | Sync. | Ours | GT-offset |
|---|---|---|---|---|---|
| 4DGS | Plenoptic | 29.52 | 30.27 | 30.84 | 30.88 |
| | Blender | 27.61 | 29.29 | 33.39 | 35.33 |
| EDGS | Plenoptic | 28.07 | 27.82 | 29.70 | 29.40 |
| | Blender | 28.86 | 31.20 | 33.35 | 38.97 |
| K-Planes | Plenoptic | 28.84 | 28.87 | 29.67 | 30.52 |
| | Blender | 30.40 | 32.46 | 32.94 | 39.32 |

Averaged over all baselines and datasets, the PSNR drop relative to the GT-offset results is **6.68%** with our method, compared to **14.16%** (original) and **11.22%** (sync). These results demonstrate that our approach substantially reduces the gap to perfect synchronization, achieving nearly half the PSNR drop of the synchronized baseline.

Table 8: Comparison of processing time between different baselines and our method.

| Baseline | Sync.(min) | Ours (min) | Relative Time |
|---|---|---|---|
| 4DGS | 100 min | 164 min | 1.64× |
| EDGS | 60 min | 70 min | 1.17× |
| K-Planes | 158 min | 198 min | 1.25× |

**Training Runtime.** We report the total training time of our method compared with the synchronized baselines in Table 8. All experiments are conducted on a single NVIDIA A100 GPU. These results show that our joint optimization introduces a moderate runtime overhead, while consistently providing significant improvements in reconstruction and synchronization quality across different baselines.

**Effectiveness of the Certainty Mask.** In addition to our original reliability mask based on motion magnitude, we introduce a certainty mask derived from SEA-RAFT(Wang et al., 2024) flow confidence. To evaluate the effectiveness of the certainty-based masking strategy, we conduct an ablation study on the Sync-NeRF Blender dataset with 0-13 frames unsynchronized. The certainty mask is derived from the intermediate per-pixel confidence output of the SEA-RAFT flow estimator, which reflects the certainty of the predicted flow. Pixels with the lowest 10% confidence are removed, filtering out regions likely to contain flow errors.

Table 9: Effectiveness of the certainty mask. The three variants correspond to: using both the certainty mask and the motion-based reliability mask, using only the reliability mask, and using only the certainty mask, respectively.

| Model | PSNR | SSIM | LPIPS |
|---|---|---|---|
| w/ certainty mask + reliability mask | **37.66** | **0.9844** | **0.0139** |
| w/ reliability mask only | 36.23 | 0.9824 | 0.0152 |
| w/ certainty mask only | 36.07 | 0.9822 | 0.0161 |

The results are summarized in Table 9. Using both the certainty mask and the motion-based reliability mask achieves the best performance. Using either mask alone consistently yields lower quality, showing that the two masks provide complementary benefits: the certainty mask suppresses flow outliers in occluded or fast-moving regions, while the reliability mask focuses on informative motion. Together, they reduce the impact of erroneous flow on synchronization and mitigate noisy supervision from unreliable flow estimates.

**Offset Regularization Term.** We further investigate the impact of the offset regularization term. We conducted an ablation study on the Blender datasets from ParticleNeRF, where the temporal offset range is $[0, 37]$. The results are summarized in Table 10. As shown in the table, a large regularization weight ($\lambda = 0.02$) significantly degrades performance. By contrast, removing the regularization ($\lambda = 0$) achieves results comparable to using a small weight ($\lambda = 0.0002$). This indicates that strong regularization restricts the model's ability to handle large temporal misalignments, whereas a relaxed or negligible regularization is essential for allowing sufficient flexibility in synchronization correction.

Table 10: Ablation on the offset regularization weight $\lambda$. Large $\lambda$ severely harms synchronization performance under large temporal shifts.

| Metric | $\lambda = 0$ | $\lambda = 0.0002$ | $\lambda = 0.02$ |
|---|---|---|---|
| PSNR | 30.59 | 30.66 | 21.73 |
| MAE | 0.126 | 0.138 | 0.524 |

**Implementation Details of the Projected Scene Flow.** For EDGS, the position of each 3D Gaussian over time $t$ is represented as a combination of its base position and dynamic basis functions, $\mathbf{x}_k(t)$. The 3D scene flow between two consecutive frames is defined as $\mathbf{x}_k(t+1) - \mathbf{x}_k(t)$. The projected scene flow on the 2D image plane for camera $i$ is represented as $\hat{f}_{i,k} = \mathbf{J}_i (\mathbf{x}_k(t+1) - \mathbf{x}_k(t))$, where $\mathbf{J}_i$ is the Jacobian of the affine approximation of the projective transformation. Contributions from multiple Gaussians to the same pixel are fused via $\alpha$-blending, $\hat{f}_i = \sum_k \alpha_k \hat{f}_{i,k} \prod_{m<k}(1 - \alpha_m)$, where $\alpha_k$ denotes the opacity of the $i$-th Gaussian.

For 4DGS, inspired by Gao et al. (2024), the projected Gaussian flow $\hat{\mathbf{f}}$ is obtained by the differentiable renderer by computing per-Gaussian pixel displacements in the image plane and combining them through $\alpha$-blending. Specifically, for the $k$-th Gaussian, its displacement is computed based on its 2D mean $\mu$ and covariance $\Sigma$ at times $t_1$ and $t_2$. The computation proceeds as follows: the query pixel position $\mathbf{x}_{t_1}$ is first mapped to the Gaussian's local canonical space, and then it is mapped back to the image plane according to the Gaussian's state at $t_2$ (where $t_2 = t_1 + 1$), yielding the predicted pixel position $\mathbf{x}_{k,t_2}$. The per-Gaussian pixel displacement is then given by:

$$\mathbf{g}_k = \mathbf{x}_{k,t_2} - \mathbf{x}_{t_1}.$$

The image-space projected scene flow for camera $i$ at pixel position $\mathbf{x}_{t_1}$ is obtained by $\alpha$-blending the per-Gaussian displacements, where $\alpha_k$ denotes the $\alpha$ value of the $k$-th Gaussian:

$$\hat{f}_i = \sum_{k=1}^{K} \left( \alpha_k \prod_{m<k} (1 - \alpha_m) \right) \mathbf{g}_k$$

For hybrid methods such as K-Planes, the 3D scene flow operator $\mathcal{F}_{3D}$ is implemented by first extracting features from six orthogonal planes of the factorized 3D feature tensor, $\mathbf{XY}, \mathbf{XZ}, \mathbf{YZ}$. These features are then processed through a lightweight MLP-based scene flow decoder, producing the 3D scene flow, denoted as $\mathcal{F}_{3D}(\mathbf{F}_t)$. Subsequently, the projected scene flow for the $i$-th camera, $\hat{f}_i$, is obtained by applying the camera's projection operator $\pi$:

$$\hat{f}_i = \pi(\mathcal{F}_{3D}(\mathbf{F}_t), \mathcal{P}_i)$$

where $\mathcal{P}_i$ is the projection matrix of the $i$-th camera.

## B  MORE VISUALIZATION RESULTS

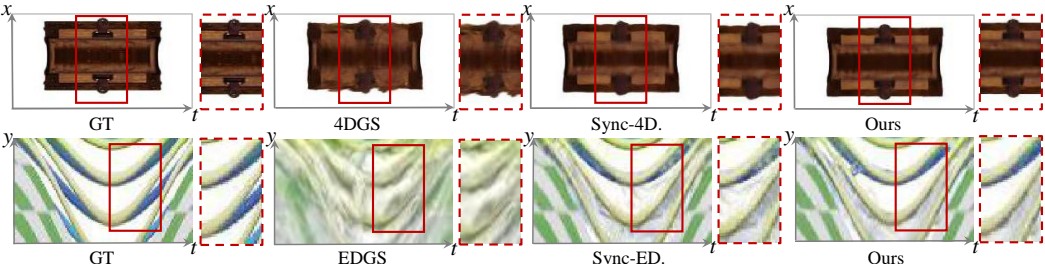

Figure 8: Visualization of spatial-temporal consistency using X-T and Y-T slices. Compared to baseline methods, our method preserves sharper temporal structures and more continuous motion patterns. Zoom-in insets highlight that our method reconstructs more coherent textures over time.

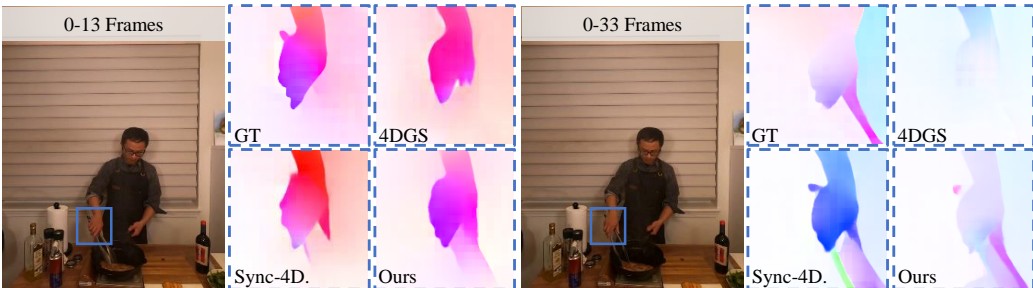

Figure 9: Comparison of optical flow predictions under increasing temporal misalignment (0–13 vs. 0–33 frames). Our method produces flow color distributions that closely resemble the ground truth (GT), preserving sharper motion boundaries and more accurate structures. In contrast, Sync-4D. and 4DGS exhibit blurred boundaries and flow color distributions that deviate significantly from the GT, especially under larger temporal offsets.

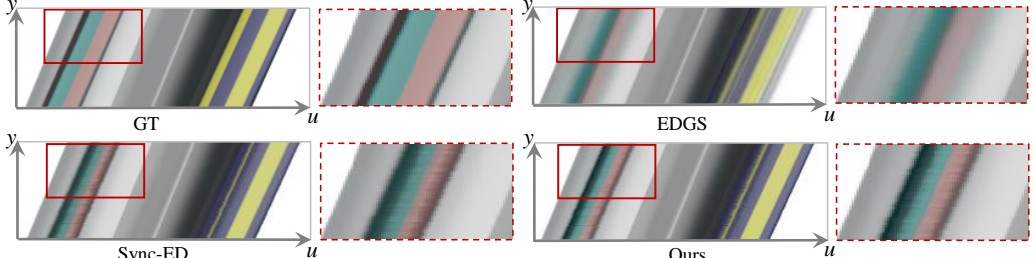

Figure 10: Comparison of spatial-temporal consistency in EPI representations ($u$-$y$ slices). Our method produces sharper and more coherent EPI lines, closely matching the ground truth, while EDGS and Sync-EDGS exhibit noticeable distortions and aliasing.

