# OpenReview forum: "DYNAMIC NOVEL VIEW SYNTHESIS FROM UNSYNCHRONIZED VIDEOS USING GLOBAL-LOCAL MOTION CONSISTENCY PRIOR"
_ICLR.cc/2026/Conference — Submitted to ICLR 2026_

### Official Review · Reviewer_3MkA · 2025-10-27

**Soundness:** 3
**Presentation:** 3
**Contribution:** 3
**Rating:** 6
**Confidence:** 4

**Summary:**

This paper aims to solve the challenge of dynamic novel view synthesis (D-NVS) from unsynchronized multi-view videos. To address the problem that existing methods easily fall into local optima in textureless scenes or with large temporal misalignments, the authors propose a "global-local motion consistency prior". The core of this method is a loss function ($\mathcal{L}_{Flow}$) that jointly optimizes the scene representation ($F_t$) and the learnable temporal offsets ($\Delta t_i$) by comparing the 2D projection of the predicted 3D scene flow ($\hat{f_i}$) with pre-computed 2D optical flow ($f_i$). The authors integrate this loss function into several mainstream D-NVS frameworks (NeRF and GS), and experiments show the method significantly reduces synchronization errors and improves reconstruction quality.

**Strengths:**

The paper presents an interesting observation: using the "anisotropy" of projected motion as a supervisory signal for temporal alignment. Compared to appearance-based photometric consistency, this prior theoretically provides a more robust signal for alignment in textureless regions. Furthermore, the method demonstrates strong generality; it is designed as a "plug-and-play" loss function that can be easily integrated into various mainstream D-NVS frameworks (including K-Planes, 4DGS, and EDGS). The experimental results are convincing, showing consistent and significant performance improvements across multiple baselines, datasets, and degrees of temporal misalignment, effectively enhancing both reconstruction quality and synchronization accuracy.

**Weaknesses:**

**W1. Dependency on Optical Flow and Limitations of the Masking Strategy:** The method's core supervision signal relies on pre-computed 2D optical flow ($f_i$). This introduces a critical external dependency, as optical flow algorithms themselves are unreliable in scenarios like occlusions, fast motion, or transparent/reflective surfaces. The authors attempt to mitigate this with a reliability mask, but this mask is based only on flow "magnitude" (selecting the top 50%), which is a coarse strategy. It ignores other major failure modes of optical flow, such as occlusion boundaries or moving textureless regions (which may have high magnitude but low accuracy), which can still introduce erroneous supervision signals into the optimization.

**W2. Missing Key Implementation Details:** The paper lacks key implementation details required for reproducibility, especially the specific implementation of the 3D flow operator ($\mathcal{F}_{3D}$) for the different frameworks (4DGS, EDGS, K-Planes). The descriptions are vague (e.g., 4DGS "implicitly realized through splatting" or EDGS "tracking... Gaussian's motions"), lacking clear mathematical formulations and differentiable implementation paths, which makes the work difficult to reproduce.

**W3. Limited Authenticity of Evaluation:** The experimental evaluation has limited authenticity. All "unsynchronized" data was created by artificially applying random offsets to already synchronized public datasets. The paper does not test on any *truly* unsynchronized, real-world multi-camera recordings. Therefore, it is unverified whether the method can generalize to handle real-world asynchrony issues such as rolling shutter, clock drift, and exposure differences.

**Questions:**

**Q1. Circular Dependency in Optimization:** This method requires jointly optimizing the scene representation ($F_t$) and the temporal offsets ($\Delta t_i$). However, in the early stages of training, both $F_t$ (which depends on $\Delta t_i$) and $\Delta t_i$ (whose supervision $\hat{f_i}$ depends on $F_t$) are likely incorrect. How does the optimization process break this dependency and ensure convergence? What prevents the optimization from collapsing into a local minimum where an incorrect $F_t$ and an incorrect $\Delta t_i$ mutually reinforce each other?

**Q2. Mismatch of Pre-computed Optical Flow:** The paper uses pre-computed optical flow $f_i$ (e.g., calculated between frames $t$ and $t+1$) as the supervision signal. However, because the videos themselves are unsynchronized (e.g., by an offset $\Delta t_i$), this $f_i$ actually represents the motion between global scene times $(t+\Delta t_i)$ and $(t+1+\Delta t_i)$. The model's objective, however, seems to be the scene flow corresponding to global time $t$. How do you handle this fundamental mismatch in the supervision signal $f_i$ caused by $\Delta t_i$ itself?

**Q3. Temporal Baseline and Scale Inconsistency:** The loss function $\mathcal{L}_{Flow}$ compares the pre-computed 2D optical flow $f_i$ and the projected 3D scene flow $\hat{f_i}$. However, $f_i$ is typically a discrete displacement field corresponding to a specific time interval (e.g., $\Delta \tau=1$ frame), while $\hat{f_i}$ (derived from $\mathcal{F}_{3D}$) might represent an instantaneous velocity or a displacement over a different time step. Please clarify how $f_i$ and $\hat{f_i}$ are kept strictly consistent in terms of their time scale and units.

**Q4. Supervision Mismatch under Large Misalignments:** When handling large temporal offsets (e.g., 0-37 frames), the pre-computed optical flow $f_i$ is still a short-term motion calculated between adjacent frames (e.g., $t$ and $t+1$). However, the $\Delta t_i$ the model needs to optimize is a long-term offset spanning many frames. How can this short-term motion supervision signal ($f_i$) effectively guide the model to find a correct, large-scale temporal offset ($\Delta t_i$)?

---

> ### Author Response · Authors · 2025-11-20
> **Response to Reviewer 3MkA(split 1 of 4)**
>
> **Weakness1/3 Dependency on Optical Flow and Limitations of the Masking Strategy**
>
> We thank the reviewer for the detailed and accurate critique. Our motivation for using optical flow supervision is that purely photometric optimization is particularly weak in textureless regions, where RGB differences provide almost no signal. Although optical flow is not perfect(especially under occlusions, fast motion, or reflective surfaces), modern learning-based estimators (e.g., PWC-Net[1], VideoFlow[2], RAFT[3] used in our framework) **generally provide a stronger motion prior than photometric cues alone**. These models are trained on large video datasets and incorporate multi-scale cost volumes, long-range correlations, and iterative refinement, making them substantially more robust than simple local matching.
>
> At the same time, we agree with the reviewer that our original mask design, which relied only on flow magnitude, is indeed coarse and does not specifically address major failure modes in ambiguous high-magnitude regions. To address this, we now incorporate **flow confidence** in addition to motion magnitude.
>
> Specifically, we adopt SEA-RAFT[4], which outputs a per-pixel certainty (confidence) value in addition to the flow. This confidence is derived from the model’s internal correlation volume and refinement iterations and reflects how reliable the predicted flow is at each location. We construct a **certainty mask** by discarding the lowest-confidence 10% of pixels. This directly removes regions where flow is likely inaccurate. In parallel, we keep our motion-based reliability mask to filter out small-motion or uninformative areas.
>
> We evaluate three variants (certainty mask only, reliability mask only, and both combined) on the Sync-NeRF Blender dataset (0–13 frame offset). Results are shown below:
>
> | Model                   | PSNR |  SSIM |  LPIPS |
> |----------------------------------------|---------|----------|-----------|
> | w/ cert. mask + rel. mask    | **37.66** | **0.9844** | **0.0139** |
> | w/ rel. mask              | 36.23 | 0.9824 | 0.0152 |
> | w/ cert. mask             | 36.07 | 0.9822 | 0.0161 |
>
> The combined design achieves the best performance, showing that confidence-based and motion-based filtering are complementary: the certainty mask suppresses flow outliers (e.g., occlusions, reflections), while the reliability mask focuses supervision on informative motion. Together, they substantially reduce the impact of erroneous flow on synchronization and directly address the reviewer’s concern about noisy supervision.
>
> In summary, we acknowledge the limitations of the original masking strategy and have incorporated a more principled **confidence-aware masking mechanism**, which significantly improves robustness. These clarifications have included in the revised manuscript.
>
>
> [1]Sun D, Yang X, Liu M Y, et al. Pwc-net: Cnns for optical flow using pyramid, warping, and cost volume[C]//Proceedings of the IEEE conference on computer vision and pattern recognition. 2018: 8934-8943.
>
> [2]Shi X, Huang Z, Bian W, et al. Videoflow: Exploiting temporal cues for multi-frame optical flow estimation[C]//Proceedings of the IEEE/CVF International Conference on Computer Vision. 2023: 12469-12480.
>
> [3]Teed Z, Deng J. Raft: Recurrent all-pairs field transforms for optical flow[C]//European conference on computer vision. Cham: Springer International Publishing, 2020: 402-419.
>
> [4]Wang Y, Lipson L, Deng J. Sea-raft: Simple, efficient, accurate raft for optical flow[C]//European Conference on Computer Vision. Cham: Springer Nature Switzerland, 2024: 36-54.

---

> > ### Author Response · Authors · 2025-11-20
> > **Response to Reviewer 3MkA(split 2 of 4)**
> >
> > **Weakness2/3 Missing Key Implementation Details**
> >
> > We thank the reviewer for pointing this out. Due to pages limitations, the description in Sec. 4.1 was not sufficiently detailed; we have provided clearer implementation details in the appendix in the revised manuscript.
> >
> > For EDGS[1], the position of each 3D Gaussian over time $t$ is represented as a combination of its base position and dynamic basis functions, $\mathbf{x} _ k(t)$. The 3D scene flow between two consecutive frames is defined as $ \mathbf{x} _ k(t+1) - \mathbf{x} _ k(t)$. The projected scene flow on the 2D image plane for camera $i$ is represented as $\hat{f}_{i,k} = \mathbf{J}_i  (\mathbf{x}_k(t+1) - \mathbf{x}_k(t))$, where $\mathbf{J}_i$ is the Jacobian of the affine approximation of the projective transformation. Contributions from multiple Gaussians to the same pixel are fused via $\alpha$-blending, $\hat{f} _ i = \sum _ k \alpha _ k \hat{f} _ {i,k} \prod _ {m<k} (1-\alpha _ m)$, where $\alpha_k$ denotes the opacity of the $i$-th Gaussian.
> >
> > For 4DGS[2], inspired by [3], the projected Gaussian flow $\hat{\mathbf{f}}$ is obtained by the differentiable renderer by computing per-Gaussian pixel displacements in the image plane and combining them through α-blending. Specifically, for the $k$-th Gaussian, its displacement is computed based on its 2D mean $\mu$ and covariance $\Sigma$ at times $t_1$ and $t_2$. The computation proceeds as follows: the query pixel position $\mathbf{x} _ {t_1}$ is first mapped to the Gaussian's local canonical space, and then it is mapped back to the image plane according to the Gaussian's state at $t_2$ (where $t_2 = t_1 + 1$), yielding the predicted pixel position $\mathbf{x}_{k,t_2}$. The per-Gaussian pixel displacement is then given by:
> >
> > $$
> > \mathbf{g} _ k= \mathbf{x} _ {k,t_2} - \mathbf{x} _ {t_1}.
> > $$
> >
> > The image-space projected scene flow for camera $i$ at pixel position $\mathbf{x}_{t_1}$ is obtained by α-blending the per-Gaussian displacements, where $\alpha_k$ denotes the α value of the $k$-th Gaussian:
> >
> > $$
> > \hat{f} _ i = \sum _ {k=1}^{K} \left( \alpha _ k \prod _ {m<k} (1 - \alpha _ m) \right) \mathbf{g} _ k $$
> >
> > For hybrid methods such as K-Planes[4], the 3D scene flow operator $\mathcal{F} _ {3D}$ is implemented by first extracting features from six orthogonal planes of the factorized 3D feature tensor, $\mathbf{XY}, \mathbf{XZ}, \mathbf{YZ}$ . These features are then processed through a lightweight MLP-based scene flow decoder, producing the 3D scene flow, denoted as $\mathcal{F}_{3D}(\mathbf{F}_t)$. Subsequently, the projected scene flow for the $i$-th camera, $\hat{f}_i$, is obtained by applying the camera’s projection operator $\pi$:
> >
> > $\hat{f} _ i = \pi(\mathcal{F} _ {3D}(\mathbf{F} _ t), \mathcal{P} _ i)$
> >
> > where $\mathcal{P}_i$ is the projection matrix of the $i$-th camera. This implementation is fully differentiable, allowing gradients to propagate through both the feature extraction and the MLP decoder.
> >
> > [1]Katsumata K, Vo D M, Nakayama H. A compact dynamic 3d gaussian representation for real-time dynamic view synthesis[C]//European Conference on Computer Vision. Cham: Springer Nature Switzerland, 2024: 394-412.
> >
> > [2]Wu G, Yi T, Fang J, et al. 4d gaussian splatting for real-time dynamic scene rendering[C]//Proceedings of the IEEE/CVF conference on computer vision and pattern recognition. 2024: 20310-20320.
> >
> > [3]Gao Q, Xu Q, Cao Z, et al. Gaussianflow: Splatting gaussian dynamics for 4d content creation[J]. arXiv preprint arXiv:2403.12365, 2024.
> >
> > [4]Fridovich-Keil S, Meanti G, Warburg F R, et al. K-planes: Explicit radiance fields in space, time, and appearance[C]//Proceedings of the IEEE/CVF Conference on Computer Vision and Pattern Recognition. 2023: 12479-12488.

---

> > > ### Author Response · Authors · 2025-11-20
> > > **Response to Reviewer 3MkA(split 3 of 4)**
> > >
> > > **Weakness3/3 Limited Authenticity of Evaluation**
> > >
> > > We appreciate the reviewer’s insightful comments regarding the authenticity of the evaluation and the potential impact of rolling shutter, clock drift, and exposure differences.
> > >
> > > **(1) Authenticity of unsynchronized data**. In our experiment, the used Plenoptic dataset is captured by real multi-camera system, as reported in the dataset’s documentation, the originally captured videos already exhibit mild inter-camera asynchrony. In our study, we further enlarge the temporal offsets on top of these realistic conditions to stress-test our method. The consistent gains we observe in synchronization accuracy and reconstruction quality on this dataset indicate that our approach is effective beyond purely synthetic timing perturbations.
> > >
> > > **(2) On rolling shutter, clock drift, and exposure differences**. Our method focuses on correcting **large frame-level** temporal offsets, which are the dominant source of inconsistency in unsynchronized multi-camera videos. In comparison, rolling-shutter readout delays and clock drift introduce much **smaller sub-frame** timing variations, whose magnitude is negligible relative to offsets on the order of tens of frames. Exposure differences mainly influence appearance rather than temporal alignment and are often handled within the reconstruction pipeline. While these finer-level effects are meaningful in broader settings, they fall outside the scope of the frame-level synchronization problem studied here and represent a natural direction for future extensions.
> > >
> > > **(3) Dataset limitations and future directions**. Current public multi-camera dynamic scene datasets (including Plenoptic) provide only the RGB videos and camera intrinsics/extrinsics; parameters necessary to model rolling shutter, long-term drift, or exposure variations are not available. Given these data constraints, it is infeasible to explicitly model these effects within our current optimization framework. In the future, we plan to build better datasets with richer camera metadata and explore synchronization and reconstruction models that explicitly account for such real-world asynchrony factors.

---

> > > > ### Author Response · Authors · 2025-11-20
> > > > **Response to Reviewer 3MkA(split 4 of 4)**
> > > >
> > > > **Q1. Circular Dependency in Optimization**
> > > >
> > > > To address the circular dependency between the scene representation and temporal offsets, we adopt a **staged optimization strategy** to ensure stable convergence. In the initial phase, the temporal offsets are frozen while only the scene representation is optimized, allowing a stable initial scene to be learned. Once the scene representation has partially converged, both the scene representation and temporal offsets are jointly optimized. At this stage, the scene representation is reliable to a certain extent, enabling the flow supervision to effectively guide offset learning. Finally, the temporal offsets are frozen again, and only the scene representation is further refined to improve reconstruction quality. This staged strategy ensures stable convergence throughout the optimization process.
> > > >
> > > > **Q2. Mismatch of Pre-computed Optical Flow**
> > > >
> > > > We thank the reviewer for raising this important point. We clarify that under our formulation, the ”mismatch” described by the reviewer does not occur.
> > > >
> > > > For each camera $cam_i$, the pre-computed optical flow $f_i$ is computed within that camera’s own video, i.e., between local frames $(t_i, t_i+1)$. We learn a single temporal offset $\Delta t_i$ that maps local time to global time $t_i + \Delta t_i$.
> > > >
> > > > Thus, the predicted 3D flow at global time $T$ is supervised by the optical flow from the same camera at its corresponding local frame(illustrated as Eqn.3):
> > > >
> > > > $\mathcal{L} _ {\text{Flow}} = \sum_{i=1}^M \| f_i(t_i) - \hat{f}_i(t_i + \Delta t_i) \|_1.$
> > > >
> > > > Equivalently, emphasizing that optical flow supervises the correct global time after alignment:
> > > >
> > > > $\mathcal{L} _ {\text{Flow}} = \sum_{i=1}^M \| f_i(t_i - \Delta t_i) - \hat{f}_i(t_i) \|_1.$
> > > >
> > > > These formulations show that each camera’s flow supervises only its own offset-corrected global timestamp, and is never required to align with other cameras’ local frames. Therefore, the mismatch described by the reviewer does not arise in our method.
> > > >
> > > > **Q3. Temporal Baseline and Scale Inconsistency**
> > > >
> > > > We thank the reviewer for the careful reading. We would like to clarify that, in our implementation, the pre-computed 2D optical flow $f_i$ and the model-predicted, projected 3D scene flow $\hat{f}_i$ are strictly consistent in terms of temporal baseline and units, both representing pixel displacements between consecutive frames $t$ and $t+1$. The pre-computed optical flow is estimated over adjacent frames ($\Delta t = 1$) using RAFT or similar algorithms. The 3D scene flow is defined as the displacement **between consecutive frames $t$ and $t+1$**, and is then projected to the image plane, ensuring both are in units of pixels per frame.
> > > >
> > > > **Q4. Supervision Mismatch under Large Misalignments**
> > > >
> > > > We thank the reviewer for raising this question. We acknowledge that the challenge does exist: as the per-camera temporal offsets increase, the overlap of valid time ranges across cameras decreases. Nevertheless, the short-term motion cues remain informative within the overlapping windows and continue to guide the recovery of large temporal offsets. Incorrect large temporal offsets lead to discrepancies between the observed optical flow and the projected scene flow over time steps, resulting in strong gradients that drive the optimization towards offsets where these short-term motions align. However, this indeed makes the optimization more difficult, as also reflected in our results (i.e. the performance drops when the temporal offset is increased).
> > > >
> > > > For future work, we plan to explore hierarchical spatio-temporal pyramids inspired by optical-flow estimation, which could allow motion supervision to be applied at multiple temporal scales. Such a pyramid would allow motion supervision to be applied progressively at multiple temporal resolutions, from coarse to fine, which can help guide the alignment even when videos exhibit very large frame-level misalignments. We will include this direction in our future work and thank the reviewer for this question.

---

> ### Author Response · Authors · 2025-11-27
> **Looking Forward to Further Feedback**
>
> Dear Reviewer,
>
> Thanks for your valuable time reviewing our paper. Your insights and suggestions have been instrumental in refining our submission, and we are deeply grateful for your time and effort. We are eager to hear your additional feedback of our paper. Thank you!

---

### Official Review · Reviewer_KFz8 · 2025-10-31

**Soundness:** 2
**Presentation:** 3
**Contribution:** 3
**Rating:** 6
**Confidence:** 3

**Summary:**

A global-local motion consistency prior and derived loss function is proposed to address novel-view synthesis in dynamic scenes with unsynchronised views. The prior is used to temporally align multi-view videos by explicitly estimating a temporal offset between views in an optimization framework for a sharper novel-view synthesis.

**Strengths:**

The method is intuitive and simple. Elegant presentation of the method, by abstracting from the instantiation of operator F. Significant quantitative and qualitative gains across all baselines and benchmarks.

**Weaknesses:**

The writing style is not uniform. The abstract is super clear, while in the introduction some sentences do not follow academic standard: like "to achieve synchronization by removing cables" and "as everyone knows, the most commonly used photometric consistency prior is prone to local minima optimization" (I personally can intuitively follow and guess but I do not know this, since i do not work on time synchronisation for NVS). Indeed, you introduce section 3.2.1 to explain this so it is not known to everyone. Sentence at line 186 needs revision. Line 217 is splitted (residual from the figure 3).

Optical flow is precomputed, it would have been interesting to see the proposed prior applied to an end-to-end trainable architecture including optical flow training. But i understand this is beyond the specific scope of this paper.

**Questions:**

No questions.

---

> ### Author Response · Authors · 2025-11-20
> **Response to Reviewer KFz8**
>
> We sincerely appreciate the reviewer’s valuable comments on our writing. Some of our original sentences indeed suffered from unclear academic expression, excessive assumed knowledge, and insufficient objectivity. We have carefully revised the paper to ensure a consistent, formal academic style.
>
> ## **Weakness 1/2 The writing style is not uniform**
>
> **Comment on the writing issue at L044**: Regarding the sentence at L044, The original phrase “to achieve synchronization by removing cables” was to emphasize that our method avoids additional hardware triggers or synchronization cables, thereby reducing both financial and labor costs. In the revised manuscript, we have replaced it with the more academically appropriate expression: “**to achieve synchronization without relying on hardware-based solutions**”.
>
> **Comment on the writing issue at L051**: For the sentence at L051, we agree that the previous phrasing was not sufficiently objective and assumed too much prior knowledge. In low-texture regions, the image intensity is nearly constant within the spatial neighborhood, making the brightness gradient close to zero; consequently, the photometric consistency loss fails to provide informative gradient signals for optimization. In repetitive-texture regions (e.g., grids or tiled floors), many visually similar candidate locations exist, causing ambiguity in correspondence and resulting in multiple plausible minima of the photometric error. These issues, namely gradient degradation and matching ambiguity, diminish the discriminative power and optimization stability of photometric consistency supervision, thereby limiting its reliability and accuracy in dynamic scene reconstruction. In the revised manuscript, we have **removed the informal phrase “as everyone knows” and added a concise version of the explanation above**.
>
> **Comment at L186 and L217**: Additionally, we modified the sentence at L186 to: We focus on correcting large frame-level temporal misalignments and do not account for minor sub-frame variations induced by other camera effects(e.g., rolling shutter, clock drift, and exposure differences). The formatting issue at L217 has also been corrected in the revised PDF.
>
> ## **Weakness2/2 About End-to-End Trainable Framework with Optical Flow**
>
>  We thank the reviewer for the insightful suggestion regarding integrating optical flow estimation into an end-to-end trainable system. This is indeed a promising direction; however, to the best of our knowledge, no existing work has jointly coupled optical flow estimation with dynamic scene reconstruction, especially under temporally misaligned multi-view settings.
>
> In our current design, we intentionally separate flow computation from the joint optimization of temporal offsets and dynamic fields, which makes our method naturally **compatible with various pre-computed optical flow algorithms**. As shown in Table 4, our framework does not rely on any particular flow estimator, and **improvements in optical-flow accuracy may further enhance the reconstruction results**.
>
> Moreover, incorporating an optical-flow network into the full reconstruction pipeline would substantially **increase training cost and reduce efficiency**, since existing flow networks are typically trained offline on large-scale datasets and used directly during inference rather than optimized jointly with the reconstruction system. We appreciate the reviewer's constructive suggestion and hope our clarification addresses this point.

---

> ### Author Response · Authors · 2025-11-27
> **Looking Forward to Further Feedback**
>
> Dear Reviewer,
>
> We truly appreciate your earlier feedback. If any questions remain unresolved, we would be happy to clarify them before the discussion period closes.
> If our responses have fully addressed your concerns, we kindly hope you may consider raising the score accordingly. We are eager to hear your additional feedback of our paper.
>
> Thank you very much for your time.

---

### Official Review · Reviewer_ya4T · 2025-11-01

**Soundness:** 3
**Presentation:** 3
**Contribution:** 2
**Rating:** 4
**Confidence:** 2

**Summary:**

This paper proposes a global-local motion consistency prior to address temporal misalignment in unsynchronized dynamic NeRF and GS frameworks. By aligning global scene flow with optical flow, the method reduces synchronization errors by ~50% and improves PSNR by up to 4 dB, outperforming prior approaches.

**Strengths:**

The paper is well-structured and easy to follow. The proposed method demonstrates strong effectiveness across different D-NVS architectures and achieves state-of-the-art performance.

**Weaknesses:**

The proposed method offers limited novelty. This work primarily adds two constraints — Flow and Offset Regularization Term — with the latter assigned a very small weight (0.0002), while the weight for Flow is not specified.

In most reconstruction tasks, temporal misalignment is typically addressed through camera synchronization or pre-alignment of captured data. It is recommended to include results under temporally pre-aligned conditions as a baseline to more clearly demonstrate the effectiveness of the proposed approach.

Please specify the weight used for the Flow term and provide an ablation study to evaluate its influence on the model’s performance.

**Questions:**

Please provide additional evidence to further demonstrate the novelty and effectiveness of the proposed method.

---

> ### Author Response · Authors · 2025-11-20
> **Response to Reviewer ya4T**
>
> We thank the reviewer for the feedback.
>
> Our contribution is not just "add constrains" but **the core scientific insight that motion consistency is a fundamentally more robust prior than photometric consistency for temporal offset optimization**. Building on this insight, we introduce a global–local motion consistency prior to guide temporal offset estimation in unsynchronized videos. We further develop a flow-based consistency loss that integrates naturally into existing dynamic reconstruction frameworks. This design effectively reduces synchronization errors and improves reconstruction quality over prior methods.
>
> **The offset regularization term.** Since our baseline Sync-NeRF includes an offset regularization term, we keep this term for consistency and assign it a very small weight in our experiments. This choice is justified by our ablation in the Offset Regularization Term section of Table 10 in the revised PDF, which shows that strong regularization actively harms performance by preventing the model from correcting large offsets.
>
> **The weight of the flow term.** For example, in 4DGS, $\lambda_{\text{Flow}}$ is set to 0.05 for real-world scenes and 0.5 for blender scenes. We additionally include an ablation study on the weight of the flow loss, conducted on the 0–13 frame unsynchronized Plenoptic Datasets using the 4DGS baseline, with results shown below:
>
> | Metric | $\lambda_{\text{Flow}}=0.1$ | **$\lambda_{\text{Flow}}=0.05$** | $\lambda_{\text{Flow}}=0.01$ | $\lambda_{\text{Flow}}=0.001$ |
> |--------|------------------------------|----------------------------------|-------------------------------|--------------------------------|
> | PSNR   | 31.05                        | **31.29**                        | 30.94                         | 30.81                          |
> | SSIM   | 0.9397                       | **0.9403**                       | 0.9402                        | 0.938                          |
> | LPIPS  | 0.0979                       | **0.0970**                        | 0.0974                        | 0.0987                         |
>
> As the results show, reducing its weight weakens the effectiveness of the flow-supervision term and negatively impacts reconstruction quality. We have added this ablation study into the revised paper.
>
> **Results of synchronized videos.** Due to page limitations, the results under temporal pre-alignment (denoted as **GT-offset**) have been placed in the appendix of our previous submission (Table 7 in Additional Quantitative Results on Synchronized Videos).
>
> | Baseline   | Dataset   | Ori.   | Sync.  | Ours   | GT-offset |
> |------------|-----------|--------|--------|--------|-----------|
> | 4DGS       | Plenoptic | 29.52  | 30.27  | 30.84  | 30.88     |
> |            | Blender   | 27.61  | 29.29  | 33.39  | 35.33     |
> | EDGS       | Plenoptic | 28.07  | 27.82  | 29.70  | 29.40     |
> |            | Blender   | 28.86  | 31.20  | 33.35  | 38.97     |
> | K-Planes   | Plenoptic | 28.84  | 28.87  | 29.67  | 30.52     |
> |            | Blender   | 30.40  | 32.46  | 32.94  | 39.32     |
>
> Averaged over all baselines and datasets, the PSNR drop relative to the GT-offset results is **6.68%** with our method, compared to **14.16%**(Ori.) and **11.22%**(Sync.). These results demonstrate that our approach substantially reduces the gap to perfect synchronization, achieving nearly **half** the PSNR drop of the sync. baseline.

---

> ### Author Response · Authors · 2025-11-27
> **Looking Forward to Further Feedback**
>
> Dear Reviewer,
>
> We sincerely appreciate your time and effort in evaluating our work. We have carefully addressed your insightful questions and hope that our responses help to better highlight the impact and results of our work. We kindly request you to review our responses and share any additional feedback or concerns if any.
>
> Thank you very much for your time.

---

### Official Review · Reviewer_dbve · 2025-11-02

**Soundness:** 2
**Presentation:** 3
**Contribution:** 2
**Rating:** 6
**Confidence:** 4

**Summary:**

This paper tackles dynamic novel view synthesis without hardware synchronization.

Task configuration
* input: multi-view videos with camera poses
* output: dynamic NeRFs or Gaussians and temporal offsets for synchronization

Solution
* global-local motion consistency prior: consistency between 3D (learnable) and 2D (pre-computed) flows
* Masking out the consistency loss on textureless region

Experiments
* Competitor: Sync-NeRF on 4DGS, EDGS, and Kplanes.
* Improvements on synchronization: The average temporal error reduces about half.
* Improvements on reconstruction: 1~4 dB

**Strengths:**

Originality:
1. Please see weakness 1

Quality:
1. The experiments thoroughly compare the competitors.

Clarity:
1. Section 3.1 clearly defines the task: reconstructing the scene and finding the temporal offsets of cameras for given a set of videos and their camera poses.
2. Section 3.2 clearly provides the intuition (consistency between 3D and 2D flows) and the method (adding the difference between projected scene flow and optical flows, masking out textureless region).

Significance:
1. The proposed method improves reconstruction (PSNR, SSIM, LPIPS) by large margins.
2. Please see weakness 3.
3. The experiments cover large offset ranges.

**Weaknesses:**

1. Section 2.2. implies “Optical flow for synchronization is not novel”. The section should explain why this paper is not obvious compared to previous papers. Especially, [Neural Scene Flow Fields for Space-Time View Synthesis of Dynamic Scenes] (should be cited) already introduces 3D-2D consistency.
2. The paper will be easier to read if the terms are more precisely named and redundant sentences are removed. E.g., L021 we develop~ is redundant with L017.  It is weird to name 3D scene flow and 2D optical flow as global and local motion, respectively. L017 will be easier as follows: To tackle this issue, we impose 3D-2D consistency loss to encourage 3D scene flow to match 2D optical flow.  L095 is redundant with L093.
3. The importance of the components of the proposed method should be explained by the ablation study. Currently, we do not know whether 3D-2D flow consistency is more important than masking. If masking is more important, the paper would better put more emphasis on the masking than 3D-2D flow consistency because 3D-2D flow consistency is not novel as in weakness 1. Furthermore, the problem statement L051 and the proposed method does not match because the limitation in textureless regions is the same for RGB and flow.

**Questions:**

1. Resolving weakness 1 will improve my rating regarding originality.
2. Resolving weakness 2 will improve my rating regarding clarity.
3. Resolving weakness 3 will improve my rating regarding significance and logical soundness.

---

> ### Author Response · Authors · 2025-11-20
> **Response to Reviewer dbve(split 1 of 2)**
>
> ## **Weakness 1/3 Clarifying Novelty over Previous Methods**
>
> We thank the reviewer for raising this important point and for highlighting the relevance of Neural Scene Flow Fields (NSFF). We agree that optical-flow-based synchronization alone, as mentioned in Section 2.2, is not a novel contribution by itself.
>
> Our intention in Section 2.2 was to emphasize that prior works in dynamic vision, such as CamLiFlow[1] which demonstrate that motion features provide strong cues and often outperform purely photometric constraints in dynamic correspondence estimation. However, CamLiFlow is built for **stereo camera** setups with **strong view overlap**, where both cameras observe nearly the same regions. Our scenario is fundamentally different: the cameras often have **minimal shared content** or even **near-360° coverage**, making stereo-style 2D correspondence estimation unreliable. In these cases, purely 2D optical-flow-based approaches break down, and their assumptions no longer hold. As a result, these methods could not be leveraged in our experiments.
>
> Our novelty is not in using optical flow, but in **how motion-derived cues are incorporated into a global-local dynamic reconstruction framework** to solve the previously unaddressed problem of **unsynchronized multi-view dynamic capture**. Our contributions focus on (1) establishing why motion features are more reliable than photometric cues for unsynchronized multi-view inputs, (2) integrating them into a reconstruction-oriented optimization pipeline rather than a pure correspondence estimation task, and (3) providing a principled explanation of how motion consistency drives robust alignment under challenging real-world asynchrony.
>
> We acknowledge that omitting NSFF[2] from the discussion was an oversight. NSFF introduces neural scene flow fields to jointly model geometry and 3D motion across time, enforcing 3D–2D flow-based consistency for dynamic view synthesis. Its formulation further validates the reliability of using motion-based 3D–2D consistency in our synchronization framework. We have added the related description to Section 2.1.
>
> [1]Liu H, Lu T, Xu Y, et al. Camliflow: bidirectional camera-lidar fusion for joint optical flow and scene flow estimation[C]//Proceedings of the IEEE/CVF conference on computer vision and pattern recognition. 2022: 5791-5801.
>
> [2]Li Z, Niklaus S, Snavely N, et al. Neural scene flow fields for space-time view synthesis of dynamic scenes[C]//Proceedings of the IEEE/CVF conference on computer vision and pattern recognition. 2021: 6498-6508.
>
> ## **Weakness 2/3 Redundancy in Terminology and Sentences**
>
> We appreciate the reviewer’s suggestions for improving clarity. In the revised manuscript, we have removed redundant sentences and enhanced overall readability.
>
> In our terminology, **"global"** corresponds to the 3D scene flow, which represents a coherent, scene-wide motion field defined in the full 3D space, with its temporal origin aligned with the global time reference. In contrast, **"local"** refers to the 2D optical flow, which is a view dependent image plane measurement specific to each individual camera. Due to the lack of synchronization among cameras, optical flow measurements from different cameras are generated on misaligned local time axes rather than a shared global timeline. The core idea of our method is to enforce **consistency between this global 3D motion field and the local 2D observations**. From our perspective, the term “global–local motion consistency” precisely captures the relationship between these two complementary motion cues.
>
> To improve clarity, we have revised the corresponding descriptions in the revised manuscript as suggested, specifically in Lines 17–21. We have also revised Lines 93–95 to address the redundancy, and we thank the reviewer for pointing this out.

---

> > ### Author Response · Authors · 2025-11-20
> > **Response to Reviewer dbve(split 2 of 2)**
> >
> > ## **Weakness 3/3 Importance of flow consistency vs. masking and Textureless regions**
> >
> > We thank the reviewer for this insightful observation, which indeed touches on the central mechanism of our method.
> >
> > **(1) Importance of 3D–2D flow consistency vs. masking**
> >
> > In our formulation, **the 3D–2D flow consistency term is the core of synchronization**, while the reliability mask is only an auxiliary filtering mechanism. Without the flow term, the mask cannot function at all.
> > To make this clearer, we added ablations on the EDGS baseline under different temporal offsets. As shown below, **removing the flow term causes a significantly larger performance drop** than removing the mask, confirming that flow consistency is the dominant component:
> >
> >  Offset      | Model                  | PSNR  | SSIM  | LPIPS |
> > |-------------|------------------------|-------|-------|-------|
> > | 0–13 Frames | Ours                   | **29.72** | **0.9344**| **0.0995**|
> > | 0–13 Frames | w/o Reliability Mask   | 29.53 | 0.9325| 0.1021|
> > | 0–13 Frames | w/o flow               | 27.62 | 0.9245| 0.1025|
> > | 0–21 Frames | Ours                   | **29.34** | **0.9297**| **0.1168**|
> > | 0–21 Frames | w/o Reliability Mask   | 28.77 | 0.9260| 0.1216|
> > | 0–21 Frames | w/o flow               | 27.80 | 0.9195| 0.1168|
> > | 0–33 Frames | Ours                   | **30.05** | **0.9331**| 0.1172|
> > | 0–33 Frames | w/o Reliability Mask   | 29.38 | 0.9294| **0.1153**|
> > | 0–33 Frames | w/o flow               | 28.03 | 0.9212| 0.1200|
> >
> > These results make it clear that 3D–2D consistency is the essential term, while the mask provides an incremental improvement by filtering unreliable regions.
> >
> > **(2) Textureless regions: why flow still helps**
> >
> > We agree that textureless regions pose challenges for both RGB-based and flow-based cues. However, the theoretical analysis in our paper focuses on demonstrating that motion features are inherently more robust than texture features for the synchronization problem. The acquisition of accurate optical flow, however, constitutes a separate research issue orthogonal to our work.
> >
> > Our motivation for using optical flow is not that flow is perfect, but that **motion cues remain informative even where color cues fail**. We only require flow to be more reliable than RGB cues. Improving flow accuracy is a valuable direction but **orthogonal to the contribution of this paper**.
> >
> >
> > To obtain motion cues that remain reliable in such challenging regions, we use modern learning-based flow estimators (e.g., RAFT[1], PWC-Net[2], VideoFlow[3], which we employ) leverage multi-scale cost volumes, long-range correlations, and temporal reasoning. These architectures are trained on large-scale datasets and are explicitly designed to handle ambiguous or low-texture regions better than per-scene photometric optimization.
> >
> > Finally, our method uses flow selectively: reliable flow contributes strong 2D-3D consistency, while unreliable regions are filtered by our mask. Thus, even in textureless regions where RGB cues provide no signal, motion cues remain a stronger supervisory source, and the mask ensures robustness when flow quality degrades.
> >
> >
> > [1]Teed Z, Deng J. Raft: Recurrent all-pairs field transforms for optical flow[C]//European conference on computer vision. Cham: Springer International Publishing, 2020: 402-419.
> >
> > [2]Sun D, Yang X, Liu M Y, et al. Pwc-net: Cnns for optical flow using pyramid, warping, and cost volume[C]//Proceedings of the IEEE conference on computer vision and pattern recognition. 2018: 8934-8943.
> >
> > [3]Shi X, Huang Z, Bian W, et al. Videoflow: Exploiting temporal cues for multi-frame optical flow estimation[C]//Proceedings of the IEEE/CVF International Conference on Computer Vision. 2023: 12469-12480.

---

> ### Author Response · Authors · 2025-11-27
> **Looking Forward to Further Feedback**
>
> Dear Reviewer,
>
> Thank you again for the time and effort you have devoted to reviewing our work. We have carefully addressed your earlier concerns in the discussion. If there are any remaining points you would like us to clarify, we would be glad to provide further details.
>
> If you feel that our responses satisfactorily resolve the concerns, we would sincerely appreciate your consideration in raising the score in your final evaluation.
>
> Thank you for your time and effort in reviewing our paper.

---

### Author Response · Authors · 2025-12-03
**Summary of Authors’ Response to Reviewers**

We sincerely thank all reviewers for their constructive and insightful feedback. We are encouraged by **their recognition of the strengths of our work:**

1.clear, and well-structured method with elegant design(R_dbve, R_ya4T, R_KFz8);

2.thorough and comprehensive experiments across large offset ranges and baselines(R_dbve, R_KFz8);

3.strong, state-of-the-art performance and significant reconstruction improvements across frameworks and datasets (R_dbve, R_ya4T, R_KFz8, R_3MkA);

4.key observation with plug-and-play generality and broad applicability (R_3MkA).

We have summarized the reviewers' main concerns below and highlighted how our revisions marked in **blue** and **clarifications have comprehensively addressed these points**.

**Clarification of Novelty and Core Contribution**

R_dbve and R_ya4T asked for clearer clarification of novelty, especially regarding optical-flow-based synchronization. We have clarified that **our novelty is not merely using optical flow or adding constraints, but the insight that motion consistency is a more robust prior than photometric cues for temporal offset estimation in unsynchronized multi-view videos**. Unlike prior 2D correspondence methods assuming stereo setups with strong overlap, our setting often involves minimal shared content. Our framework has aligned 3D scene motion with 2D observations, and with a flow-based consistency loss, effectively reduced synchronization errors and improved reconstruction quality across architectures and datasets.

**Terminology and Methodological Details**

R_dbve and R_ya4T raised writing issues, including redundancy and unclear phrasing, while R_3MkA noted missing implementation details. We have clarified terminology, standardized style, removed redundancies, and added concise flow computation details in the appendix, addressing their concerns on methodology, terminology, and clarity.

**Ablation and Empirical Validation**

R_dbve questioned the role of the flow term and mask and noted that low-texture regions are inherently ambiguous for both RGB and flow; R_ya4T asked about regularization and flow-loss weights and performance under pre-alignment; R_3MkA raised concerns about our masking strategy.

Our ablations have shown that **3D–2D flow consistency is the dominant synchronization cue, with the mask providing only auxiliary stability (updated Table 3)**. For textureless regions, we have clarified to R_dbve that motion cues are inherently more reliable than RGB for synchronization and that perfect optical flow estimation is orthogonal to our contribution. Using modern learning-based flow and a reliability mask, our method has still obtained effective motion supervision where RGB cues fail while remaining robust to imperfect flow.

Ablations requested by R_ya4T (updated Table 5, Table 10) have confirmed that **strong regularization prevents recovering large offsets and that reducing flow-loss weight weakens synchronization**; results on pre-aligned videos (Appendix Table 7 from the **initial submission**) have further shown that our method **markedly narrows the gap to perfect synchronization, addressing prior concerns that could influence reviewer ratings on method effectiveness**.

To address R_3MkA, we have **augmented our mask with SEA-RAFT flow certainty**, and Appendix Table 9 has shown this significantly improves robustness. Overall, these ablations have validated that motion-driven 3D–2D consistency is key to reliable synchronization and reconstruction, addressing the reviewers’ prior concerns.

**Authenticity of Evaluation and Flow Supervision**

R_3MkA raised concerns about the authenticity of evaluation on real unsync data, real-world effects like rolling shutter, and methodological details of flow supervision. We have clarified that the real-world dataset we used was captured by a real multi-camera system with mild inherent asynchrony, which we further enlarged and still observed consistent gains. Rolling shutter, clock drift, and exposure affect sub-frame or appearance-level aspects, beyond our frame-level synchronization focus, and cannot be modeled with current public datasets. We also have clarified flow supervision functions and relevant training details. For future work, we plan to leverage multi-scale spatio-temporal pyramids to handle larger temporal misalignments.


**Conclusion**

Overall, the reviewers’ feedback has confirmed the novelty, clarity, and effectiveness of our approach, while pointing out areas for clarification, additional implementation details, and expanded ablations. We have revised the manuscript to address all these points, resulting in clearer presentation, more comprehensive validation, and stronger evidence for the robustness of our method.

---

### Meta-Review · Area_Chair_WPgf · 2025-12-15

**Summary:**

The reviewers raised several concerns regarding the novelty, methodology, experimental design, and presentation of the paper. Reviewer dbve and Reviewer ya4T questioned the novelty of the proposed approach, noting that the use of optical flow and 3D–2D motion consistency has been explored in prior work on dynamic scene reconstruction, such as Neural Scene Flow Fields (NSFF). From their perspective, the main contribution—enforcing consistency between projected 3D scene flow and pre-computed 2D optical flow—appears to be an incremental regularization rather than a fundamentally new methodological contribution.

Reviewer 3MkA raised concerns about the optimization formulation, particularly the potential circular dependency between temporal offset estimation and scene geometry or scene flow reconstruction. The reviewer noted that inaccurate early-stage estimates could lead to unreliable supervision for time offsets, potentially causing convergence to suboptimal solutions, and that the paper does not sufficiently justify the stability of this joint optimization process.

Concerns were also raised regarding the experimental evaluation. Reviewer dbve pointed out that the unsynchronized data used in the experiments are generated by artificially introducing frame offsets to synchronized datasets, rather than being captured under real-world unsynchronized conditions. As a result, it remains unclear how well the proposed method would generalize to more complex real-world scenarios involving temporal drift or other capture-related artifacts.
In addition, Reviewer KFz8 and Reviewer dbve commented on issues related to clarity and presentation, including inconsistent terminology, non-academic phrasing in parts of the introduction, and minor formatting or typographical errors.

Overall, while the paper demonstrates quantitative improvements on standard metrics such as PSNR, the reviewers collectively felt that the limited novelty, unresolved concerns regarding optimization stability, and the lack of validation on real-world unsynchronized data reduce the overall impact of the work.

**Reviewer Concerns:**

**Addressed**

Reviewer dbve, Reviewer ya4T, and Reviewer 3MkA highlighted the lack of ablation studies, including the impact of the flow term, masking strategy, and offset regularization, as well as the absence of a pre-aligned baseline. In response, the authors provided the requested ablation experiments, which clarify the contribution of each component and demonstrate the importance of the proposed 3D–2D flow consistency.

Reviewer 3MkA raised concerns regarding the potential mismatch introduced by using pre-computed optical flow under temporal misalignment. The authors addressed this issue by clarifying how the flow supervision remains valid despite temporal offsets and by explaining the consistency assumptions underlying the optimization.

Reviewer dbve expressed concerns about the effectiveness of the method in textureless regions. The authors clarified that while such regions are ambiguous for both RGB- and flow-based cues, motion information provides inherently stronger and more discriminative signals than photometric cues for temporal synchronization.


**Outstanding**

Reviewer dbve and Reviewer ya4T raised concerns regarding the novelty of the paper. While the authors clarified that their motion consistency serves as a more robust prior than photometric cues specifically for temporal offset estimation in challenging unsynchronized multi-view settings, the reviewers remained unconvinced due to the existence of prior flow-based methods addressing related problems.

Reviewer KFz8 and Reviewer dbve commented on issues of writing clarity and presentation. Although some revisions were made, parts of the manuscript still contain awkward phrasing and inconsistent terminology.

Reviewer 3MkA expressed concerns about the circular dependency in jointly optimizing scene flow and temporal offsets. The authors’ rebuttal primarily clarified implementation and training details, but did not provide a theoretical guarantee or formal analysis demonstrating the stability of this joint optimization.

Reviewer 3MkA and Reviewer dbve also raised concerns about the limited real-world validation. The authors acknowledged this limitation, noting that their evaluation focuses on frame-level synchronization and that real-world effects such as rolling shutter, clock drift, and exposure variations are beyond the scope of the current work and cannot be modeled with existing public datasets.

**Reviewer Scores:**

Reviewer dbve: 6, Change to 4, as the novelty concern was not resolved.

Reviewer ya4T: 4, No change, as the novelty concern was not resolved.

Reviewer KFz8: 6, No change, as the writing issues were largely addressed, but the reviewer’s confidence remained moderate (score 3).

Reviewer 3MkA: 6, change to 4. This reviewer raised critical theoretical questions regarding circular dependency, and the authors’ rebuttal failed to provide a theoretical guarantee.

---

### Decision · Program_Chairs · 2026-01-26

Reject